# Evaluation of a Two-Tier Screening Pathway for Congenital Adrenal Hyperplasia in the New South Wales Newborn Screening Programme

**DOI:** 10.3390/ijns6030063

**Published:** 2020-08-12

**Authors:** Fei Lai, Shubha Srinivasan, Veronica Wiley

**Affiliations:** 1Department of NSW Newborn Screening Programme, The Sydney Children Hospital Network, Westmead, NSW 2145, Australia; veronica.wiley@health.nsw.gov.au; 2Faculty of Medicine and Health, The University of Sydney Children’s Hospital Westmead Clinical School, Westmead, NSW 2145, Australia; shubha.srinivasan@health.nsw.gov.au; 3Department of Endocrinology, The Sydney Children’s Hospital Network, Westmead, NSW 2145, Australia

**Keywords:** congenital adrenal hyperplasia, newborn screening, 17-α hydroxyprogesterone, immunoassay, liquid chromatography tandem mass spectrometry, screening pathway

## Abstract

In Australia, all newborns born in New South Wales (NSW) and the Australia Capital Territory (ACT) have been offered screening for rare congenital conditions through the NSW Newborn Screening Programme since 1964. Following the development of the Australian Newborn Bloodspot Screening National Policy Framework, screening for congenital adrenal hyperplasia (CAH) was included in May 2018. As part of the assessment for addition of CAH, the national working group recommended a two-tier screening protocol determining 17α-hydroxyprogesterone (17OHP) concentration by immunoassay followed by steroid profile. A total of 202,960 newborns were screened from the 1 May 2018 to the 30 April 2020. A threshold level of 17OHP from first tier immunoassay over 22 nmol/L and/or top 2% of the daily assay was further tested using liquid chromatography tandem mass spectrometry (LC-MS/MS) steroid profiling for 17OHP (MS17OHP), androstenedione (A4) and cortisol. Samples with a ratio of (MS17OHP + A4)/cortisol > 2 and MS17OHP > 200 nmol/L were considered as presumptive positive. These newborns were referred for clinical review with a request for diagnostic testing and a confirmatory repeat dried blood spot (DBS). There were 10 newborns diagnosed with CAH, (9 newborns with salt wasting CAH). So far, no known false negatives have been notified, and the protocol has a sensitivity of 100%, specificity of 99.9% and a positive predictive value of 71.4%. All confirmed cases commenced treatment by day 11, with none reported as having an adrenal crisis by the start of treatment.

## 1. Introduction

Congenital adrenal hyperplasia (CAH) is an autosomal recessive disorder that occurs when there is a disruption in any of the enzymes along the adrenal steroidogenesis pathway [1,2,3]. CAH is categorised depending on which enzyme is affected. The severity of symptoms is inversely correlated with nonfunctioning enzyme activity [4]. The most common enzyme defect is 21-hydroxylase deficiency accounting for over 95% of cases. This form of CAH is subtyped into classical CAH and non-classical CAH. Classical CAH is further divided into salt-wasting CAH (SWCAH) and simple-virilising CAH (SVCAH) [4,5,6]. SWCAH, accounting for approximately 75% of classical CAH presentation, is the most severe form of CAH [7,8]. The incidence worldwide of classical CAH is usually considered to be approximately 1:14,000 to 1:18,000; however, it varies depending on the ethnic background [2]. Reported observed incidence is highest in Yupik Eskimos from Southern Alaska at 1:282 [9].

Newborn screening for CAH began with the development of a radioimmunoassay by Pang et al., 1977 [10] measuring 17 α-hydroxyprogesterone(17OHP) using blood on microfilter paper. Since then, worldwide CAH screening or pilot studies have ensued [11,12,13,14,15,16,17,18,19,20,21,22]. Newborn screening for CAH is aimed at identifying newborns with SWCAH promptly to prevent a life threatening adrenal crisis, thus reducing morbidity and mortality in affected individuals [23]. Detrimental adrenal and salt wasting crises occur within the first 2 to 3 weeks of life in newborns with SWCAH. Early clinical symptoms can be non-specific such as poor feeding, vomiting, diarrhoea and sepsis, which can lead to erroneous diagnoses [24]. Population newborn screening provides the opportunity to detect and treat those with CAH before the onset of significant symptoms.

Newborn bloodspot screening in Australia has been established since the late 1960s [25,26,27]. There are five state government-funded programs, which are located in Adelaide, Brisbane, Melbourne, Perth, and Sydney [28]. However, until recently New Zealand was the only center in Australasia screening for CAH [28]. In Australia, in New South Wales (NSW) and the Australia Capital Territory (ACT), a two-year pilot study was performed from 1st October 1995 to 30th September 1997, assessing the benefits and feasibility of screening CAH in newborns using an immunoassay for 17OHP with different follow-up action depending on birthweight and concentration of 17OHP compared to clinical diagnoses for newborns born in other states of Australia. Based on the findings from this study, implementing screening for CAH was considered justified [29]. However, funding for screening for CAH was not approved by the state governments. The decision was based on the concern of the harm caused by the number of false positive cases in the screened population as well as the evidence that the diagnosis for unsuspected cases in the screened population (median age: 13 days) was not significantly less than for the unscreened population (median age: 16 days).

In Australia, the inclusion of newborn screening for CAH was proposed to each state government at various times over the intervening years by the Australasian Pediatric Endocrine Group and the Human Genetic Society of Australasia (HGSA) [30]. A request was forwarded to the federal health minister in 2013. However, the process of adding CAH screening was challenging, as there was an absence of clear national policies or guidelines endorsed by all governments to support uniform newborn bloodspot screening. CAH was included as a recommended disorder in Australia in May 2018 due to the efforts of a time-limited multi-disciplinary CAH Assessment Working Group (CAHWG). The CAHWG also trialed the “Newborn Screening Bloodspot National Policy Framework”, which included the tools for assessment of the inclusion or removal of recommended conditions [25].

The NSW Newborn Screening Programme commenced screening for CAH in May 2018 using the proposed recommended two-tier method protocol. This included all dried blood spot (DBS) samples being initially measured for 17-α hydroxyprogesterone (17OHP) using immunoassay followed by a second tier of steroid profiling using liquid chromatography tandem mass spectrometry (LC-MS/MS) for a percentage of samples with the highest 17OHP level. Whilst the CAHWG noted that the percentage could have differed in each state program, it was estimated to have been between 1 and 2% of sample results. This paper provides an evaluation of the first 2 years of implementation of screening for CAH in NSW.

## 2. Materials and Methods

### 2.1. Samples

All parents are provided with a multimedia information on newborn screening, including a pamphlet (Newborn Bloodspot Screening-Tests to Protect Your Baby) by the maternity health provider. Furthermore, educational videos and specific fact sheets are available on the website for parents and health professionals (https://www.schn.health.nsw.gov.au/find-a-service/laboratory-services/newborn-screening).

Newborn screening is not mandatory in Australia, and therefore following parent(s) consent, a heel prick blood spot sample is collected onto special pre-printed filter card provided by NSW Newborn Screening Programme, ideally when the baby is 48 to 72 h after birth. The sample is air dried before being sent to the laboratory via a courier or local Australia Post. All DBS samples received in the laboratory by 10:15 am each day are processed as a batch on that working day.

Once received in the laboratory, the integrity and validity of each sample is determined. A repeat DBS sample is requested for samples that are deemed unsuitable due to being contaminated, insufficient or collected less than 24 h after birth, or having been collected after blood products were given to the newborn. A repeat sample is also requested at 1 month of age for any low birth weight (<1.5 kg) or premature (<30 weeks gestation) newborn. DBS samples with relevant clinical or family history information are processed for all routine screening tests plus assessed for further testing inclusion. Each initial DBS sample is allocated a unique laboratory sample identification number and with its corresponding demographic information entered into the laboratory information system (LabMaster Database) where a unique patient identification number is generated. Repeat samples received are matched with the previously generated unique patient identification number.

The DBS samples are then punched into 96 well microtiter plates using Panthera-Puncher^TM^ 9 (Perkin Elmer, Turku, Finland) to simultaneously punch and distribute 3.2 mm blood disc into 6 different microtiter plates, one of which is a plate for immunoassay of 17αOHP.

### 2.2. Immunoassay

The concentration of 17OHP is initially determined on all DBS samples received using GSP^®^ Neonatal 17 α-OH-Progesterone assay kit (PerkinElmer, Turku, Finland) on the 2021-0010 Genetic Screening Platform^®^ (PerkinElmer, Turku, Finland) (GSP). The GSP assay is a competitive dissociation-enhanced lanthanide fluorescent immunoassay (DELFIA) (PerkinElmer, Turku, Finland). The kit provides the antibody-coated microtiter plates, calibrators, quality controls and all the reagents required to perform the immunoassay. A set of external quality control samples is also included in each daily assay. In our laboratory, the validation study for this kit using 5000 deidentified routine samples from full-term, normal birthweight infants established the 98th centile for 17OHP was 21.8 nmol/L whole blood. Therefore, samples with 17OHP ≥22 nmol/L and/or falling in the top 2% of all samples received for the daily assay for any birthweight were further tested using the second-tier LC-MS/MS steroid panel analysis. Samples with any clinical information or family history relevant to CAH were also tested using the LC-MS/MS steroid panel analysis.

### 2.3. LC-MS/MS

Unlabeled 17OHP, hydrocortisone and 4-Androstene-3,17-dione (A4) were obtained from LGC Dr. Ehrenstorfer GmbH (Augsburg, Germany). The isotopically labeled internal standard D8-17-OHP, [17-Hydroxyprogesterone(2,2,4,6,6,21,21,21-D8, 98%)], D7-Androstenedione, [4-Androstene-3,17-dione (2,2,4,6,6,16,16-D7, 97%)] and D4-Cortisol [Cortisol (9,11,12,123-D4, 98%)] (Cambridge Isotope Laboratories, Inc, Andover, MA, USA). Deep well microtiter plates (1000 µL) were supplied from LVL Technologies (Crailsheim, Germany), and microplate 96 plate pp flat bottom were supplied from Greiner Bio-One International (Frickenhausen, Germany). Fisher Chemical™ Optima™ LC-MS solvent from Thermo Fisher Scientific Australia Pty Ltd. (Victoria, Australia) and formic acid were obtained from Ajax Univar, Thermo Fisher Scientific (Victoria, Australia). Ammonium acetate suitable for mass spectrometry was supplied from Sigma Aldrich (WGK, Darmstadt, Germany).

The LC-MS/MS steroid assay for quantitation of 17OHP (MS17OHP), A4 and cortisol was a modified version of the assay from Rossi et al. [31]. The modifications were that stock for 17OHP, A4 and cortisol, and were made by dissolving unlabeled solid steroid compound in methanol:isopropanol (80:20 *v*/*v*) instead of ethanol; 75 µL of each calibrator was spotted onto filter cards rather than 25 µL; only one 3.2 mm blood disc was punched from each calibrator, control and sample and eluted with 220 µL of methanol:water (95:5 *v*/*v*) containing internal standards (deuterated 17OHP, A4 and cortisol) in a deep well polypropylene plate (1000 µL). The eluate was transferred to a 96 well flat bottom polypropylene plate and dried using warm air (40 °C) and reconstituted with 200 µL of methanol:water (50:50 *v*/*v*) containing 2 nM ammonium acetate and 0.1% formic acid. Samples were analysed using ACQUITY XEVO^®^TQ-S (Waters, Milford, MA, USA). The ACQUITY XEVO TQS Targetlynx™ software (version 4.1, Waters, Milford, MA, USA) was used to calculate each steroid concentration. Results of steroid quantitation for a daily batch were available within 2 h. In our laboratory, the assay has proven to be robust and reproducible with a linear calibration curve (r^2^ = 0.99) for all three steroid analytes and a coefficient variation of <10% for each of the three steroid analytes.

### 2.4. Criteria for and Follow-Up of an Abnormal Screen

A combination of the MS17OHP concentration and the ratio (17OHP + A4)/cortisol) was used to determine whether diagnostic testing was required. Newborns with a MS17OHP level >200 nmol/L or >25 nmol/L with a ratio of >2 were considered screen positive for CAH and referred for diagnostic testing. The pediatrician or general practitioner named on the DBS card was contacted to organize urgent further samples, including a repeat DBS sample and plasma, to quantitate electrolytes, glucose and a full steroid profile and to perform a clinical review of the baby. Figure 1 shows the analytical protocol. Results obtained by the screening laboratory were reviewed together with the diagnostic results and clinical evaluation to determine the diagnosis.

## 3. Results

There were 202,960 newborns tested during the period of May 2018 to April 2020, including 102,865 males. Of those screened, 2308 were from infants with very low birth weight (<1.5 kg). There were 206,469 samples, including the repeat dried blood spot collection (e.g., for screen positive, initial sample unsuitable or due to very low birthweight) analysed for 17OHP level using immunoassay. All samples were analysed for 17OHP level before day 8 of age with the exception of 0.02% of newborns with low birth weight and 0.22% of newborns with normal birth weight. Second-tier LC-MS/MS steroid profiling was required for 4218 samples after selecting the top 2% threshold of the daily immunoassay and any screen positive samples arising from the immunoassay. Of the total number from these samples, 927 (40.2%) newborns with very low birth weight and 2441 (1.2%) newborns with normal birthweight had a 17OHP level above 22 nmol/L.

Data collected showed that the 17OHP concentration from both immunoassay and LC-MS/MS obtained from newborns with very a low birth weight tended to be higher than that of the newborns with normal birth weight. Refer to Figure 2.

The number of presumptive positive samples based solely on MS17OHP > 25 nmol/L was 241; however, after applying the ratio calculation of (MS17OHP + A4)/cortisol or for samples with MS17OHP > 200 nmol/L, there were 14 newborns who were deemed presumptive positive. Following diagnostic sample results and clinical review, 10 were proven to have CAH with 9 SWCAH and 1 newborn classified SVCAH. There was a higher proportion of males in the newborns (6/10) diagnosed with CAH.

Samples with clinical information supplied such as ambiguous genitalia, indeterminate sex and query CAH were also tested with LC-MS/MS steroid profiling regardless of the immunoassay 17OHP level. A total of 20 newborns had clinical information suggestive of CAH. Out of the 20 newborns, 17 newborns had an initial immunoassay result for 17OHP of less than 13 nmol/L and normal MS17OHP level. The other 3 newborns (case numbers 1, 5 and 7 in Table 1 and Table 2) were confirmed to have CAH. All confirmed cases were initially notified by day 9 of life, and all had treatment commenced by day 20.

## 4. Discussion

Despite inclusion of screening for CAH in many newborn screening programs internationally, it has not been included in all developed programs. There remain reservations, as CAH can be detected through clinical assessment and there are noted to be high false-positive rates generated from immunoassay in low birth weight premature infants [22,32,33]. A study from the United Kingdom argued that screening for CAH has no impact on the morbidity and mortality of patients with CAH and therefore does not include CAH in its screening program [34]. It has also been suggested that newborn screening for CAH only benefits male newborns, as females can be clinically detected due to virilisation [35]. However, countries around the world screening for CAH have demonstrated that the benefits from early detection of newborns with CAH include reducing morbidity and mortality, especially for newborns with SWCAH, and can assist in gender assignment for newborn with SVCAH [36].

There have been several strategies implemented by screening programs to improve the specificity of CAH screening. Improvement in the specificity was observed when the cut-off level of 17OHP was stratified based on either gestational age or birth weight and/or age of sampling [1,14,16,18,37,38,39,40]. Although gestational age stratification of 17OHP concentration was shown to give higher specificity, birth weight stratification has been more widely used [22,41,42]. However, even with the implementation of these strategies, 1% of newborn may require recollection [40]. Similarly, in NSW a two-year pilot program from 1st October 1995 to 30th September 1997 [29] showed that despite stratified action limits for low birth weight newborns there were 6% of infants <2 kg birth weight requiring further sample collection compared to 0.3% of infants with birth weight >2 kg. In an effort to simplify the potential use of multiple action limits and assess the expected total workload for each state program, the CAHWG recommended the use of a percentile cut-off for referral to second tier without stratification due to birth weight or gestational age [43]. By performing a second-tier assay on the top 2% of the daily samples received, our laboratory screening of ~100,000 newborns per year was an average required to test 8 samples by steroid profile each day, which could have been reduced to 5 to 6 depending on the stratified action limits. The time difference and cost for processing 8 versus 5 samples was deemed insignificant.

The use of a second-tier LC-MS/MS steroid profile was first presented by Lacey et al. measuring 17OHP, A4 and cortisol [24]. By incorporating a second tier of steroid profiling using LC-MS/MS and the use of (MS17OHP + A4)/cortisol) ratio, the NSW Newborn Screening Programme has successfully screened over 200,000 newborns for CAH. In order to simplify the test cascade algorithm, it was determined that 2% of the daily population of samples would be tested with the second-tier assay. During initial evaluation of the 17OHP immunoassay on 5000 samples, the 98th centile of those with a birth weight >1.5 kg was 22 nmol/L whole blood. Therefore, to ensure all samples received that would be in the top 2% of a year, all samples with 17OHP >22 nmol/L were further tested. Using this protocol 14 newborns required further samples. There were 10 newborns diagnosed with CAH after diagnostic testing and a full clinical review: 1 SVCAH; 9 SWCAH (5 males, 3 females and 1 indetermine sex (chromosomally female)). All 9 infants with SWCAH had no prior family history of CAH, although 3 (cases 1, 5 and 7) of the newborns did have clinical observations noted on the DBS sample (Table 1 and Table 2). The newborn with SVCAH (case 7) had a family history of CAH. Although this newborn had a steroid ratio of 1, the MS17OHP level was grossly elevated, prompting further follow-up action.

The four presumptive positive newborns requiring additional follow-up due to abnormal ratio were all deemed to be normal after either a DBS sample recollection, clinical review or diagnostic testing, and all remain well. Two of the newborns were extremely premature (Table 1). The protocol used therefore only provided a few false positives (4/202,960) and had no known false-negative results. We also successfully notified likely CAH cases before any of the newborns presented with an adrenal crisis.

There have been various studies investigating the feasibility of increasing the number of analytes in the steroid panel to increase specificity and sensitivity [44,45,46,47,48]. There are studies that show that the inclusion of 21-deoxycortisol and 11-deoxycortisol is more specific for detecting SWCAH by excluding β-hydroxylase deficiency [47,48,49,50]. Investigation of additional ratios (i.e., 17OHP + 21-deoxycortisol/cortisol) has been shown to be specific for 21 hydroxylase deficiency [49]. Further studies will be carried out to determine if the addition of 21-deoxycortisol and 11-deoxycortisol will be beneficial to our screening program.

SWCAH can present with a life-threatening adrenal crisis within the first two weeks of life [51]. Screening for CAH, notification of suspicion and diagnosis needs to be achieved before a potential adrenal crisis occurs. Using the two-tiered protocol, all suspected cases of CAH were notified by day 9 of life.

The gene that encodes the 21-hydroxylase enzyme is CYP21A2. Molecular analysis of CYP21A2 variant is hampered by the difficulty of isolating the highly homologous pseudogene CYP21A1P from the active CYP21A2 gene. Further, current molecular assays require at least 2 days to provide results, therefore the length of time to generate results is a deterrent for newborn screening [36,52]. However, variant analysis of the CYP21A2 gene has the potential for further increasing the specificity and sensitivity for screening CAH by basing it on genotype/phenotype studies. Biochemical analysis interferences, such as prematurity or stress of newborns and assay steroid cross reactivity, do not affect molecular variant analysis [53]. Variant analysis has been reported to be able to further discriminate SVCAH from SWCAH [53]; however, so far the literature suggests that variant analysis has only been used as an adjunct for screening [36]. This may change in the near future as technologies advance, for example, using next-generation sequencing for the CYP21A2 gene was reported to be cost effective and less time consuming [54]. Therefore, the emergence of targeted next-generation sequencing should be explored as a feasible screening option.

In conclusion, by following the recommended screening pathway from the national newborn bloodspot policy, the NSW Newborn Screening Programme has successfully screened over 200,000 newborns for CAH detecting an incidence of SWCAH of 1:22,551. We achieved a 100% sensitivity and a specificity of 99.9%, and the positive predictive value was 71.4%. All newborns screened with positive SWCAH were notified before any adrenal crisis occurred, thereby reducing the need for intensive care intervention. 

## Figures and Tables

**Figure 1 IJNS-06-00063-f001:**
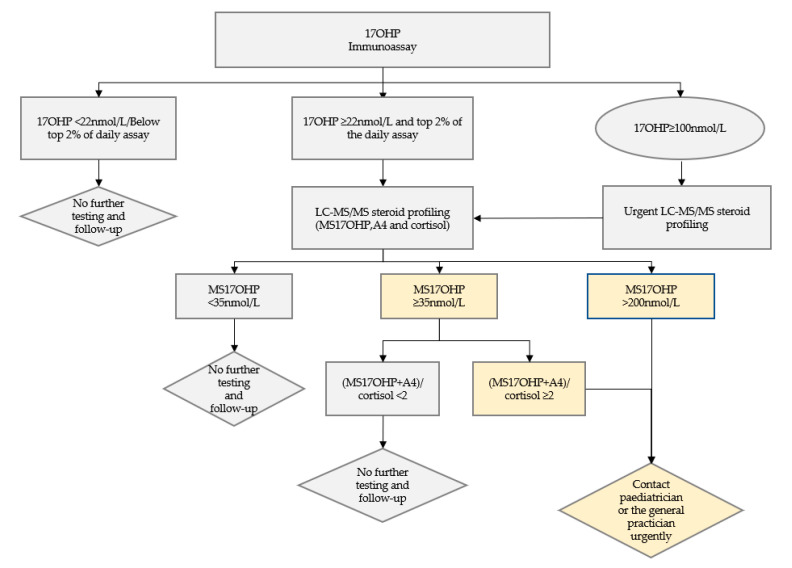
Analytical two-tier congenital adrenal hyperplasia (CAH) screening protocol using immunoassay as first tier followed by steroid profiling using liquid chromatography tandem mass spectrometry (LC-MS/MS) as a second-tier testing.

**Figure 2 IJNS-06-00063-f002:**
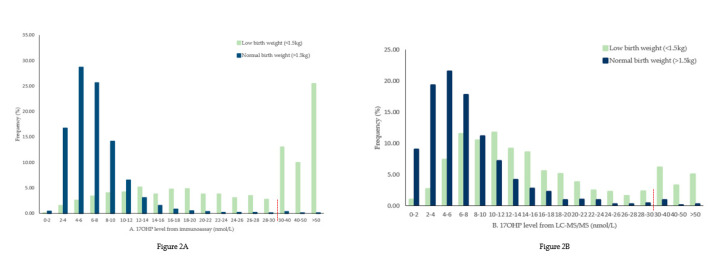
Distribution frequency of 17α-hydroxyprogesterone (17OHP) level. (**A**). 17OHP level (nmol/L) from immunoassay and (**B**). 17OHP level from LC-MS/MS.

**Table 1 IJNS-06-00063-t001:** Cases of CAH follow-up due to presumptive positive for New South Wales (NSW) Newborn Screening Programme from 1 May 2018 to 31 December 2019.

						Immuno-Assay	Results from Steroid Profiling				
Case number	Sex	Birth Weight (kg)	Gestational Age (Days)	Initial DBS Sample Collection (*)	Initial DBS Sample Received Date (*)	17OHP	MS17OHP	A4	CORTISOL	Ratio	Initial Day of Notification (*)
1	I	3.51	287	2	5	>220	97	46	47	3	5
2	F	3.3	280	2	4	>220	>250	150	18	>22	4
3	F/tw2	1.66	238	2	7	>220	173	11	73	3	9
4	M	4.37	284	2	6	>220	228	148	23	16	6
5	F	2.91	266	2	6	>220	208	64	49	6	6
6	M	1.58	216	3	6	>220	403	30	31	14	6
7	M	3.97	266	2	4	>220	455	139	810	1	4
8	M	1.58	280	3	6	>220	403	30	31	14	6
9	F	0.68	175	2	5	100.3	104	51	63	2	5
10	M	4.5	287	3	8	>220	136	49	8	23	8
11	M	4.35	280	3	6	>220	364	297	26	25	6
12	F	2.55	238	2	6	90.7	46.1	7.1	22.7	2	6
13	M	2.03	252	2	7	45	34.8	4	10.5	4	7
14	M	0.49	175	3	6	180	55.2	13	24	3	6

Analytes are displayed in nmol/L whole blood; NFT-no further follow-up; (*) all samples collection and received date are calculated from date of birth, case number 3 is a female and twin number 2.

**Table 2 IJNS-06-00063-t002:** Diagnostic results.

Case Number	Na *	K *	Glucose *	17OHP **	A4 **	CORTISOL **	TESTOSTERONE **	Family History	Symptoms	Diagnosis Suspected before Notification	Final Diagnosis	Treatment Commencement Day (*)
1	141	4.9		234		320	5.8	N	virilisation	Y	SW CAH	5
2	119	7.3	3.8	680	130	61		N	poor weight gain	N	SW CAH	11
3	130	6.1	30.3					N		N	SW CAH	9
4	136	5.8	4.2	212	>40	86	26.7	N		N	SW CAH	7
5	132	8	4.6	175	25	87	2.2	N	hypotension (associated with acute respiratory illness) virilisation	Y	SW CAH	7
6	133	5.7	3.8	>460	>37	104	51.4	N	poor feeding, preterm	N	SW CAH	8
7	136	5.3		652		88		Y	mild scrotal-transient, excess pigmentation	Y	SV CAH	2
8	133	5.7	3.8	>460	>37	104	51	N	poor feeding, preterm	N	SW CAH	8
9											NFT	
10	135	5.5	5	340	>38	31	3.5	N		N	SW CAH	10
11	132	7		498		64		N	lethargy	N	SW CAH	20
12	135	5.6	6.5	13		258					NFT	
13											NFT	
14											NFT	

* Analytes are displayed in mmol/L; ** analytes are displayed in nmol/L; case numbers 9, 13 and 14 only had a repeat dried blood spot recollection, and were clinically reviewed but had no plasma sample recollection; NFT-no further testing or follow-up; (*) days calculated from date of birth.

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
