# Peer review of "Evaluation of a Two-Tier Screening Pathway for Congenital Adrenal Hyperplasia in the New South Wales Newborn Screening Programme"

_2409-515X, 2020, doi:10.3390/ijns6030063_

Round 1
Reviewer 1 Report
In this manuscript the authors provide an evaluation of the first 2 years of experience in screening for CAH using immunoassay as first tier for 17OHP, followed by LC-MS/MS steroid profiling as a second-tier testing. The paper is well structured, the title clearly describes the content of the paper, the abstract provides a concise and complete summary. Therefore, I only have minor suggestions to be taken in consideration for improving the manuscript.
Abstract
- Line 14: the NSW acronym is written before the sentence to which it refers
- Line 23: DB is not specified
Introduction
- Line 56-57: I suggest to rewrite the following sentence: “This decision was because cases were rapidly diagnosed after clinical presentation and due to the number of false positive cases mostly in low birth weight premature newborns.”
- Line 66: authors write sometimes Newborn Bloodspot Screening, and other times newborn bloodspot screening, please standardize the style
Materials and Methods
- As assessed by the authors, the LC-MS/MS steroid method was developed in-house. Therefore, the second-tier methodology needs to be improved with more explanation. The authors do not specify the extraction procedure as well as other methodological and method development information. Analytical details of the method are totally missing. As an example: no information about DBS calibrators and QCs preparation. No information about the extraction solvent. There are not sufficient information to replicate the method. Different manuscripts have already been reported for LC-MS/MS steroid profiling for CAH diagnosis on DBS samples (PMID: 26891684, PMID: 21288182, PMID: 30623618, PMID: 14656905). If the authors refer to previous published methods, they may add some references. Otherwise, they should report methodology details in Supplementary Materials.
- As stated, it is an in-house LC-MS/MS method, again no information suggesting their method is robust or reproducible. Thus, the authors may add validation details and analytical performances of their LC-MS/MS method in Supplementary Materials.
Table 1A
- Some information is missing e.g. what is the meaning of tw2? What is the meaning of the symbol “ * ” in table 1A???
Discussion
- In the Discussion section, when demonstrating the benefits of screening for CAH and commenting on the time and on the difficulties of molecular diagnosis, they should better emphasize how the determination of a steroid panel by LC–MS/MS in the diagnostic confirmation of CAH is also informative for the following molecular testing confirmation, as reported in a recent paper (PMID: 31766536).
Author Response
Response to Reviewer 1 Comments
Point 1: Line 14: the NSW acronym is written before the sentence to which it refers
Response 1: This sentence has been restructured.
In Australia all babies born in New South Wales (NSW) and the Australia Capital Territory (ACT), have been offered screening for rare congenital conditions through the NSW Newborn Screening Programme since 1964.
Point 2: Line 23: DB is not specified
Response 2: DB is expanded to dried blood spot samples
These newborns were referred for clinical review with a request for diagnostic testing and a confirmatory repeat dried blood spot sample.
Point 3: Line 56-57: I suggest to rewrite the following sentence: “This decision was because cases were rapidly diagnosed after clinical presentation and due to the number of false positive cases mostly in low birth weight premature newborns.”
Response 3: We have now rewritten for clarification as below
The decision was based on the concern of the harm caused by the number of false positive cases in the screened population as well as the evidence that the diagnosis for unsuspected cases in the screened population (median age: 13 days) was not significantly less than for the unscreened population (median age: 16 days).
Point 4: Line 66: authors write sometimes Newborn Bloodspot Screening, and other times newborn bloodspot screening, please standardize the style
Response 4:” Capital case is used for naming otherwise in lower case and is standardised.
Point 5:
- As assessed by the authors, the LC-MS/MS steroid method was developed in-house. Therefore, the second-tier methodology needs to be improved with more explanation. The authors do not specify the extraction procedure as well as other methodological and method development information. Analytical details of the method are totally missing. As an example: no information about DBS calibrators and QCs preparation. No information about the extraction solvent. There are not sufficient information to replicate the method. Different manuscripts have already been reported for LC-MS/MS steroid profiling for CAH diagnosis on DBS samples (PMID: 26891684, PMID: 21288182, PMID: 30623618, PMID: 14656905). If the authors refer to previous published methods, they may add some references. Otherwise, they should report methodology details in Supplementary Materials.
- As stated, it is an in-house LC-MS/MS method, again no information suggesting their method is robust or reproducible. Thus, the authors may add validation details and analytical performances of their LC-MS/MS method in Supplementary Materials.
Response 5 : The development of a LC-MS/ assay was commenced on site before the publication by Rossi et al with the aid of one of the co-authors. However, the assay was modified after this publication.
We have clarified the method as below
The LC-MS/MS steroid assay for quantitation of 17OHP (MS17OHP), A4 and cortisol was a modified version of the assay from Rossi et al [32]. The modifications were that stock for 17OHP, A4 and cortisol were made by dissolving unlabelled solid steroid compound in methanol:isopropanol (80:20v/v) instead of ethanol; 75µL of each calibrator was spotted onto filter cards rather than 25µL; only one 3.2mm blood disc was punched from each calibrator, control and sample and eluted with 220µL of methanol:water (95:5v/v) containing internal standards (deuterated 17OHP,A4 and cortisol) in a deep well polypropylene plate (1000µL). The eluate was transferred to a 96 well flat bottom polypropylene plate and dried using warm air (40oC) and reconstituted with 200µL of methanol:water (50:50v/v) containing 2nM ammonium acetate and 0.1%formic acid. Samples were analysed using ACQUITY XEVO®TQ-S (Waters, Milford, MA, USA). The ACQUITY XEVO TQS Targetlynx ™ software was used to calculate each steroid concentration. Results of steroid quantitation for a daily batch were available within 2 hours. In our laboratory the assay has proven to be robust and reproducible with a linear calibration curve (r2=0.99) for all three steroid analytes and a coefficient variation of <10% for each of the three steroid analytes. Point 6:
Table 1A
- Some information is missing e.g. what is the meaning of tw2? What is the meaning of the symbol “ * ” in table 1A???
Response 6: Key for this table has been expanded
All analytes are displayed in nmol/L whole blood, NFT- no further follow up, (*) all sample collection and received date are calculated from date of birth. Case number 3 is a female and twin number 2. Case number 5 had a repeat DBS sample recollection after treatment.
Point 7:Discussion
- In the Discussion section, when demonstrating the benefits of screening for CAH and commenting on the time and on the difficulties of molecular diagnosis, they should better emphasize how the determination of a steroid panel by LC–MS/MS in the diagnostic confirmation of CAH is also informative for the following molecular testing confirmation, as reported in a recent paper (PMID: 31766536).
Response 7: this point has been expanded. Please see below
Variant analysis has been reported to be able to further discriminate SVCAH from SWCAH [54] However, so far the literature suggests that variant analysis has only been used as an adjunct for screening [37]. This may change in the near future as technologies advance, for example, using next generation sequencing for the CYP21A2 gene was reported to be cost effective and less time consuming [55] Therefore, the emergence of targeted next generation sequencing should be explored as a feasible screening option.
Reviewer 2 Report
Evaluation of a two-tier screening Pathway for Congenital Adrenal Hyperplasia in New South Wales Newborn Screening Programme
Introduction:
A large portion of the introduction focuses on the process for adding CAH to the newborn screening panel of diseases/conditions in Australia. While interesting and informative, this section does not align with your intended focus of the manuscript. Would consider whether this level of detail is necessary to include. (lines 59-81). Or maybe only highlight the specific reasons why it wasn’t universally adopted when first proposed, that align with your manuscript focus (ie…false positives, low positive predictive value).
Materials and Methods:
Your manuscript provides three different figures (1, 2, 3,) on the pre-analytical, analytical, and post-analytical process. Given the focus of your paper on the evaluation of a two-tier screening pathway for CAH, I am wondering whether you need to include the pre-analytical (figure 1) and post-analytical (figure 3) figures? I would consider what added benefit is provided by inclusion of these figures? If there is something unique about the pre- and post- analytical process, you might want to highlight this in the discussion. Additionally, the written section 2.4 titled post-analytical protocol is very short and not very meaningful to the reader (lines 149-151).
Results:
I had several questions regarding your results and algorithm:
- Because it is common practice to stratify cutoff values by either birth weight or gestational age, I am wondering why you chose to have a single cutoff value for the first tier 17OHP immunoassay? You have a very large number of samples to evaluate by second tier (>2000 for one year). Stratification would help reduce workload and throughput.
- Can you clarify in the manuscript that the top 2% of the daily immunoassay (line 172) is obtained from within the normal population (specimens with 17OHP <22nmol/L)? That is my assumption, but I am not certain I read it correctly?
- You identified 14 newborns with abnormal first and second tier test results, but only 10 babies were affected with CAH. One baby (of the 10 confirmed cases) had a normal ratio. Given that you are suggesting a screening algorithm, can you provide a new cutoff for the ratio OR else provide a cutoff for when 17OHP is markedly elevated, warranting follow-up, even if the ratio is normal? (line 237-240) If you did implement a cutoff for when MS17OHP is grossly elevated (regardless of the ratio), does this increase the total number of babies to be referred for confirmatory testing? Would there be more than 14 babies?
- You state that 20 babies had clinical information suggestive of CAH, with 3 of those confirmed. Did those cases undergo newborn screening for CAH? If yes, what were their first and second tier test results for CAH? (Line 184-185). Are these 3 cases included in the 10 total cases identified? I assume the other 17 cases had normal first or second tier results?
- In the legend for table 1B, please include the definition for NFT, similar to table 1A legend.
Discussion:
Please expand upon why you chose to NOT stratify by birth weight or gestational age. How does this decision impact your workload? Time, resources, instrument availability for second tier testing, etc… This would be a nice addition to the paragraph on outcomes of stratification (lines 222-229).
Again, please clarify how specimens were selected for second tier analysis (17OHP >= 22nmol/L AND the top 2% of the specimens with 17OHP <22nmol/L)? (Lines 233-235)
Author Response
Response to Reviewer 2 Comments
Point 1:
Introduction:
A large portion of the introduction focuses on the process for adding CAH to the newborn screening panel of diseases/conditions in Australia. While interesting and informative, this section does not align with your intended focus of the manuscript. Would consider whether this level of detail is necessary to include. (lines 59-81). Or maybe only highlight the specific reasons why it wasn’t universally adopted when first proposed, that align with your manuscript focus (ie…false positives, low positive predictive value).
Response 1: This paragraph (lines 59-81) of the introduction was intended to highlight the difficulty of the addition of CAH screening in Australia. The decision for the addition to screen for CAH for in Australia was not just impeded by the false positives, low positive predictive value of the immunoassay. There was also a lack of uniformity of decision making in terms of inclusion and exclusion of any disorders to the various state based screening programs. We accept the reviewer’s comments and have modified as below.
In Australia, the inclusion of newborn screening for CAH was proposed to each state government at various times over the intervening years by the Australasian Pediatric Endocrine Group and the Human Genetic Society of Australasia (HGSA) [30]. A request was forwarded to the federal health minister in 2013. However, the process of adding CAH screening was challenging as there was an absence of clear national policies or guidelines endorsed by all governments to support uniform newborn bloodspot screening. CAH was included as a recommended disorder to be offered for routine newborn bloodspot screening to all babies in Australia in May 2018 due to the efforts of a time-limited multi-disciplinary CAH Assessment Working Group (CAHWG). The CAHWG also trialed the “Newborn Screening Bloodspot National Policy Framework” which included the tools for assessment of the inclusion or removal of recommended conditions.
Point 2:
Your manuscript provides three different figures (1, 2, 3,) on the pre-analytical, analytical, and post-analytical process. Given the focus of your paper on the evaluation of a two-tier screening pathway for CAH, I am wondering whether you need to include the pre-analytical (figure 1) and post-analytical (figure 3) figures? I would consider what added benefit is provided by inclusion of these figures? If there is something unique about the pre- and post- analytical process, you might want to highlight this in the discussion. Additionally, the written section 2.4 titled post-analytical protocol is very short and not very meaningful to the reader (lines 149-151).
Response 2: We were considering the whole pathway from the time the baby is born to the time the baby received the treatment. We have now modified and removed the 2 extra figures.
Please see line 84-189
Point 3: Results:
I had several questions regarding your results and algorithm:
Because it is common practice to stratify cutoff values by either birth weight or gestational age, I am wondering why you chose to have a single cutoff value for the first tier 17OHP immunoassay? You have a very large number of samples to evaluate by second tier (>2000 for one year). Stratification would help reduce workload and throughput.
Response 3: The working group tasked with evaluation of the addition of CAH also recommended the screening pathway. There have therefore been modifications to the text including the final paragraph of the introduction
The NSW Newborn Screening Programme commenced screening for CAH in May 2018 using the proposed recommended two-tier method protocol. This included all dried blood spot (DBS) samples being initially measured for 17-α hydroxyprogesterone (17OHP) using immunoassay followed by a second tier of steroid profiling using liquid chromatography tandem mass spectrometry (LC-MS/MS) for a percentage of samples with the highest 17OHP level. Whilst the CAHWG noted that the percentage could differ in each state program it was estimated to be between 1 to 2% of sample results. This paper provides an evaluation of the first 2 years of implementation of screening for CAH in NSW.
As well, modification has been made to the discussion
In an effort to simplify the potential use of multiple action limits and assessing the expected total workload for each state program, the CAHWG recommended the use of a percentile cut-off for referral to second tier without stratification due to birth weight or gestational age [31]. By performing a second tier assay on the top 2% of the daily samples received, our laboratory screening ~100,000 babies per year was on average required to test 8 samples by steroid profile each day which could have been reduced to 5 or 6 depending on the stratified action limits. The time difference for processing 8 versus 5 samples was deemed insignificant.
Point 4: Can you clarify in the manuscript that the top 2% of the daily immunoassay (line 172) is obtained from within the normal population (specimens with 17OHP <22nmol/L)? That is my assumption, but I am not certain I read it correctly?
Response 4: No, the top 2% of the daily immunoassay is obtained from the daily population immunoassay. Depending on the daily 170HP population level, some days the selection of the top 2% might include samples over 22nmol/L, some days the top 2% threshold might be lower than 22nmol/L. Further detail has been included in section 2.2 of the materials and methods
In our laboratory, the validation study for this kit using 5000 deidentified routine samples from full-term, normal birthweight infants established the 98th centile for 17OHP was 21.8nmol/L whole blood. Therefore, samples with 17OHP ≥22nmol/L and/or falling in the top 2% of all samples received for the daily assay for any birthweight were further tested using the second-tier LC-MS/MS steroid panel analysis.
Point 5: You identified 14 newborns with abnormal first and second tier test results, but only 10 babies were affected with CAH. One baby (of the 10 confirmed cases) had a normal ratio. Given that you are suggesting a screening algorithm, can you provide a new cutoff for the ratio OR else provide a cutoff for when 17OHP is markedly elevated, warranting follow-up, even if the ratio is normal? (line 237-240)
Response 5: Further details have been added
Case 7 flagged our attention as there was clinical information noted on the dried blood spot screening card. A cutoff is provided. Please see modification below.
Newborns with a MS17OHP level >200nmol/L or >25nmol/L with a ratio of >2 were considered screen positive for CAH and referred for diagnostic testing.
Point 6: If you did implement a cutoff for when MS17OHP is grossly elevated (regardless of the ratio), does this increase the total number of babies to be referred for confirmatory testing? Would there be more than 14 babies?
Response 6: please see response 5. Using a cutoff of 200nmol/L provided no additional cases.
Point 7: You state that 20 babies had clinical information suggestive of CAH, with 3 of those confirmed. Did those cases undergo newborn screening for CAH? If yes, what were their first and second tier test results for CAH? (Line 184-185). Are these 3 cases included in the 10 total cases identified? I assume the other 17 cases had normal first or second tier results?
Response 7: Further details supplied.
All 20 babies went through the normal algorithm. 17 of the babies had 17OHP level well below 22nmol/L. Steroid profiling was also performed on these babies and the results were unremarkable. The 3(case1,5 and 7) that are confirmed are included into the 10 total identified cases.
Out of the 20 newborns, 17 newborns had an initial immunoassay result for 17OHP of less than 13nmol/L and normal MS17OHP level. The other 3 newborns (case numbers 1, 5 and 7 – Table1) were confirmed to have CAH.
Point 8: In the legend for table 1B, please include the definition for NFT, similar to table 1A legend.
Response 8: key for legend expanded.
NFT-no further testing or follow-up
Point 9: Discussion:
Please expand upon why you chose to NOT stratify by birth weight or gestational age. How does this decision impact your workload? Time, resources, instrument availability for second tier testing, etc… This would be a nice addition to the paragraph on outcomes of stratification (lines 222-229).
Response 9: Further details added. There is a bias of sample from the top2% towards low birth weight newborns.
In an effort to simplify the potential use of multiple action limits and assessing the expected total workload for each state program, the CAHWG recommended the use of a percentile cut-off for referral to second tier without stratification due to birth weight or gestational age [31]. By performing a second tier assay on the top 2% of the daily samples received, our laboratory screening ~100,000 babies per year was on average required to test 8 samples by steroid profile each day which could have been reduced to 5 or 6 depending on the stratified action limits. The time difference and cost for processing 8 versus 5 samples was deemed insignificant.
Point 10: Again, please clarify how specimens were selected for second tier analysis (17OHP >= 22nmol/L AND the top 2% of the specimens with 17OHP <22nmol/L)? (Lines 233-235)
Response 10: We use our database to electronically select the top 2% of the daily population samples. The selection of the top 2 % is similar to the algorithm used in cystic fibrosis screening.
Round 2
Reviewer 2 Report
Thank you for addressing the comments highlighted in the initial review. I have three additional recommendations.
1.) Line 175: No further follow-up was required if the ratio was <2. I think this statement is in error. You will perform a follow-up action if the 17OHP >200, regardless of the ratio value. Correct?
2.) You may want to consider a separate heading for paragraph 174-182. The paragraph is currently in the section titled LC-MSMS. The preceding paragraph has technical details on the second tier assay and instrumentation. The paragraph (lines 174-182) could be better highlighted, if it had a different title “Criteria for and follow-up of an abnormal screen” (or something similar). This is a really important paragraph and I wouldn’t want it to be missed.
3.)You state that second tier LC-MSMS was required for 4218 samples (Line 208), but then you state in line 213-214 that 927 samples were from newborns had low birth weight and 2441 samples from newborns with normal birth weight. Was the remaining 850 samples repeat screens? I think it would be helpful to reword these section to provide clarity for the reader (Lines 203-214).
Author Response
Response to Reviewer 2 Comments
Point 1: Line 175: No further follow-up was required if the ratio was <2. I think this statement is in error. You will perform a follow-up action if the 17OHP >200, regardless of the ratio value. Correct?
Response 1: the sentence have been removed.
Point 2: You may want to consider a separate heading for paragraph 174-182. The paragraph is currently in the section titled LC-MSMS. The preceding paragraph has technical details on the second tier assay and instrumentation. The paragraph (lines 174-182) could be better highlighted, if it had a different title “Criteria for and follow-up of an abnormal screen” (or something similar). This is a really important paragraph and I wouldn’t want it to be missed.
Response 2: an additional subheading had been added. Please see below.
2.4 Criteria for and follow-up of an abnormal screen
Point 3.You state that second tier LC-MSMS was required for 4218 samples (Line 208), but then you state in line 213-214 that 927 samples were from newborns had low birth weight and 2441 samples from newborns with normal birth weight. Was the remaining 850 samples repeat screens? I think it would be helpful to reword these section to provide clarity for the reader (Lines 203-214).
Response 3. The sentence and paragraph have been restructured
There were 202960 newborns tested during the period of May 2018 to April 2020 with 102865 males. Of those screened, 2308 were from infants with very low birth weight (<1.5kg). There were 206469 samples including the repeat dried blood spot collection (e.g. for screen positive, initial sample unsuitable or due to very low birthweight) analysed for 17OHP level using immunoassay. All samples were analysed for 17OHP level before day 8 of age with the exception of 0.02% of newborns with low birth weight and 0.22% of newborns with normal birth weight. Second tier LC-MS/MS steroid profiling was required for 4218 samples after selecting the top 2% threshold of the daily immunoassay and any screen positive samples arising from the immunoassay. Of the total number from these samples, 927 (40.2%) newborns with very low birth weight and 2441 (1.2%) newborns with normal birthweight had a 17OHP level above 22nmol/L.